# Estimating the Arc Length of the Optimal ROC Curve and Lower Bounding the Maximal AUC

**Song Liu**[*]
School of Mathematics
University of Bristol
Bristol, BS8 1UG, UK
song.liu@bristol.ac.uk

## Abstract

In this paper, we show the arc length of the optimal ROC curve is an $f$-divergence. By leveraging this result, we express the arc length using a variational objective and estimate it accurately using positive and negative samples. We show this estimator has a non-parametric convergence rate $O_p(n^{-\beta/4})$ ($\beta \in (0, 1]$ depends on the smoothness). Using the same technique, we show the surface area between the optimal ROC curve and the diagonal can be expressed via a similar variational objective. These new insights lead to a novel classification procedure that maximizes an approximate lower bound of the maximal AUC. Experiments on CIFAR-10 datasets show the proposed two-step procedure achieves good AUC performance in imbalanced binary classification tasks.

## 1 Introduction

The study of Receiver operating characteristic (ROC) curves has a long history in medicine [26], psychology [15] and radiology [16, 13]. In machine learning, ROC curves have been primarily used to analyze the performance of different classification algorithms [8, 9]. Indeed, the Area Under the Curve (AUC) encodes a classifier's ranking accuracy, making it a preferable performance metric for imbalanced class classification [8, 3]. In recent years, ROC curves have also been used in comparing two distributions and achieved promising results. Examples include analyzing the mode collapsing issue of Generative Adversarial nets (GAN) [25], and diagnosing the performance of an amortized Markov Chain Monte Carlo [18].

In applications that require computing statistical discrepancy between distributions (e.g. GAN [14] or Variational Inference (VI) [1]), $f$-divergences are widely used discrepancy measures. The family of $f$-divergences includes Kullback-Leibler divergence [23] and Total Variation distance. It has been shown that $f$-divergences, generally, can be expressed via variational objectives and efficiently approximated from empirical samples [29, 30].

Since the ROC curves are used as performance metrics in many two sample applications, are they in any way related to $f$-divergences? For example, can AUC be an $f$-divergence between positive and negative data distributions given some classification score function? An earlier investigation proves that the answer is no when the score function is the likelihood ratio [31]. Nonetheless, this result inspired us to look for $f$-divergence from other geometries of the ROC curve.

In this paper, we show that, when using the likelihood ratio score, a novel $f$-divergence arises from the *arc length* of the corresponding ROC curve. By leveraging this result, we can express the arc length using a variational objective and approximate it using only samples from two distributions.

---

[*]Webpage: https://allmodelsarewrong.net

36th Conference on Neural Information Processing Systems (NeurIPS 2022).

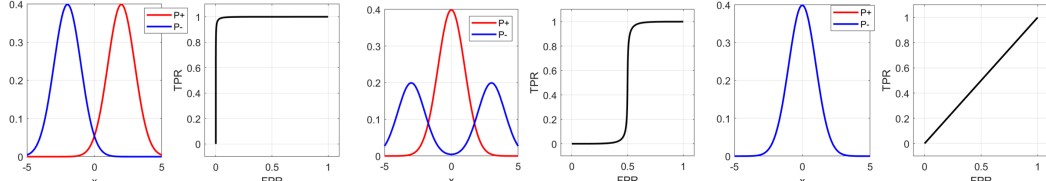

(a) different distributions, long ROC (b) different distributions, long ROC   (c) same distribution, short ROC

Figure 1: ROC curves generated for one dimensional datasets using the identity classification score function $t(x) = x$. Notice that the arc length of ROC curves seem to be a good indication on how different the positive (red) and negative (blue) data distributions are.

We show this arc length estimator is also a consistent estimator to the arctangent of likelihood ratio and has a non-parametric convergence rate $O_p(n^{-\beta/4})$, where $\beta \in (0, 1]$ depends on the smoothness of the true arctangent likelihood ratio. Moreover, by parameterizing the ROC curves of positive and negative *mixtures* distributions, the surface area between the optimal ROC curve and the diagonal can be expressed via a similar variational objective. With the help of our arctangent ratio estimator, we can approximately maximize a lower bound to this surface area. We point out the similarity between this lower bound maximization and the classic AUC maximization [3]. We show our approximated optimal score achieves comparable performance to a state of the art AUC maximizer in an imbalanced classification problem on CIFAR-10 dataset.

## 2   Background

### 2.1   An Illustrative Example

ROC curves are frequently used to visualize binary classification performance, and we often rely on the Area Under the Curve (AUC) as a numerical metric for selecting a good classifier. In this section, we highlight an often overlooked ROC geometry: arc length. We illustrate its potential as a good discrepancy measure between positive and negative data distributions.

Let us consider the one-dimensional distributions and the ROC curves in Figure 1. In this example, ROC curves are generated using the identity score function $t(x) = x$.

In both (a) and (b), $p_+$ and $p_-$ are quite different, and thus the discrepancies between positive and negative distributions should be large in both cases. In (c), the densities $p_+$ and $p_-$ are the same and consequently their discrepancy should be smaller than that of both (a) and (b). Further, notice that both (a) and (b) have long ROC curves ($\approx 2$) while (c) has a shorter one ($\approx \sqrt{2}$). This example suggests that the more similar $p_+$ and $p_-$ are, the shorter the ROC curve is.

However, we can see that $t(x) = x$ is a special choice: If $t(x) = 0$, the arc length will not reflect any discrepancy between data distributions at all. This observation inspires the following questions: Why is the arc length in this example good at telling the differences between two data distributions? Are there other score functions whose ROC arc lengths are also good discrepancy measures? What are the practical applications of studying the arc length of ROC? In the following sections, we study the arc length of a ROC curve under a probabilistic framework and provide answers to these questions.

### 2.2   ROC Curve in a Probabilistic Setting

Suppose we have positive and negative datasets $X_+ := \{\boldsymbol{x}_i^+\}_{i=1}^{n_+}$ and $X_- := \{\boldsymbol{x}_i^-\}_{i=1}^{n_-}$ drawn from two distributions $\mathbb{P}_+$ and $\mathbb{P}_-$ respectively. These distributions have respective probability density functions $p_+(\boldsymbol{x})$ and $p_-(\boldsymbol{x})$ that are both defined on the domain $\mathcal{X} \subseteq \mathbb{R}^d$. A classification score function (score function for short) takes a sample $\boldsymbol{x}$ as input and outputs a real-valued score. Suppose we have a score function $t(\boldsymbol{x}) \in \mathbb{R}$. We classify $\boldsymbol{x}$ as positive if $t(\boldsymbol{x}) > \tau$, where $\tau$ is a threshold.

Let $F_+$ and $F_-$ denote the Cumulative Distribution Functions (CDFs) of $t(\boldsymbol{x}^+)$ and $t(\boldsymbol{x}^-)$ respectively. Then we can define the following quantities:

- False Positive Rate (FPR) at threshold $\tau$, $\tilde{F}_-(\tau) := 1 - F_-(\tau)$
- True Positive Rate (TPR) at threshold $\tau$, $\tilde{F}_+(\tau) := 1 - F_+(\tau)$
- The ROC curve of a score function $t$: the graph of function $\tilde{F}_+[\tilde{F}_-^{-1}(s)]$, where $s \in [0, 1]$.

The above definition of ROC curve requires $F_-$ to be strictly increasing. In this paper, we assume $F_+$, $F_-$ to be both strictly increasing. Obviously, both $F_+$ and $F_-$ depend on the choice of score function $t$.

## 2.3 Arc Length of ROC Curve

Due to the strict monotonicity of $F_+$ and $F_-$, $\tilde{F}_+$ and $\tilde{F}_-$ form a bijective parameterization of the ROC curve in the sense that each point on this ROC curve can be written as $(\tilde{F}_-(\tau_0), \tilde{F}_+(\tau_0))$ for a unique $\tau_0 \in \mathbb{R}$. Using the line integral formula, the arc length of an ROC curve for a fixed score function $t$ can be expressed using the derivatives of $\tilde{F}_-$ and $\tilde{F}_+$:

$$\widehat{\mathrm{ROC}}(t) := \int_{-\infty}^{\infty} \sqrt{\left[\partial_\tau \tilde{F}_+(\tau)\right]^2 + \left[\partial_\tau \tilde{F}_-(\tau)\right]^2} \, \mathrm{d}\tau = \int_{-\infty}^{\infty} \sqrt{f_+(\tau)^2 + f_-(\tau)^2} \, \mathrm{d}\tau, \quad (1)$$

where $f_+(\tau)$ and $f_-(\tau)$ are the density functions of $t(\boldsymbol{x}^+)$ and $t(\boldsymbol{x}^-)$ respectively. Although (1) is an elementary result, it has been seldom discussed in the ROC literature. Authors in [5, 6] have proposed a performance metric computed over the "ROC hypersurface" and (1) is used to justify such a metric in a binary classification setting.

Using (1), we can confirm a simple geometric fact whose proof can be found in Appendix A:

**Proposition 1.** $\widehat{\mathrm{ROC}}(t) \in [\sqrt{2}, 2]$, *for all $t$.*

This result reflects the geometric observation that any monotone curve (such as ROC curve) starts and ends at two opposite corners of the ROC space $[0, 1]^2$ has an arc length between $\sqrt{2}$ and 2.

# 3 $f$-divergences Arising from ROC Arc Length

## 3.1 $f$-divergence of Score Distributions

Among many discrepancies measures, $f$-divergence has been widely used in many applications.

**Definition 1.** *Let $p$ and $q$ be densities of two continuous distributions. An $f$-divergence is defined as:* $\mathrm{D}_g(p|q) := \mathbb{E}_q \left[ g\left(\frac{p(\boldsymbol{z})}{q(\boldsymbol{z})}\right) \right]$, *where $g$ is convex and lower-semicontinuous satisfying $g(1) = 0$.*

Now let us slightly rewrite (1). Assuming $f_-$ is strictly positive (in which case $F_-$ is strictly increasing), we can write

$$\widehat{\mathrm{ROC}}(t) - \sqrt{2} = \mathbb{E}_{f_-} \sqrt{\left[\frac{f_+(\tau)}{f_-(\tau)}\right]^2 + 1} - \sqrt{2}. = \mathbb{E}_{f_-}\left[g\left(\frac{f_+(\tau)}{f_-(\tau)}\right)\right], \quad (2)$$

where $g(s) = \sqrt{s^2 + 1} - \sqrt{2}$. Equation (2) yields the first important result of this paper: $\widehat{\mathrm{ROC}}(t) - \sqrt{2}$ is an $f$-divergence between score densities $f_+$ and $f_-$ since $g(s)$ is a convex function and $g(1) = 0$. This result confirms, for any given $t$, $\widehat{\mathrm{ROC}}(t)$ is a good discrepancy for measuring positive and negative scores (i.e., score distributions). It also explains why the ROC arc length in Figure 1 is a good discrepancy measure: Given $t(x) = x$, the score distributions are same as the data distributions in each plot. The arc length of ROCs in Figure 1 plots are $f$-divergences of the score distributions hence are also $f$-divergences of data distributions.

Although $\widehat{\mathrm{ROC}}(t) - \sqrt{2}$ is an $f$-divergence of score distributions, it is not an $f$-divergence of the positive and negative *data* distributions for general $t$. The choice $t(x) = x$ in the toy example does not have simple analogues for higher dimensional datasets. Are there choices of $t$ such that the arc lengths of their ROC curves are good discrepancy measures on data distribution? In what follows, we show when $t$ is a bijective transform of the likelihood ratio $\frac{p_+(\boldsymbol{x})}{p_-(\boldsymbol{x})}$, $\widehat{\mathrm{ROC}}(t)$ encodes the differences between $p_+(\boldsymbol{x})$ and $p_-(\boldsymbol{x})$ in the form of an $f$-divergence between $p_+(\boldsymbol{x})$ and $p_-(\boldsymbol{x})$.

### 3.2 $f$-divergences of Data Distributions

Using the law of the unconscious statistician, we can express $\widehat{\mathrm{ROC}}(t)$ in terms of an expectation with respect to the negative data density $p_-(\boldsymbol{x})$: $\mathbb{E}_{p_-}\sqrt{\left[\frac{f_+(t(\boldsymbol{x}))}{f_-(t(\boldsymbol{x}))}\right]^2+1}$. Consider a special family of score functions: $t^*(\boldsymbol{x}) = \gamma\left(\frac{p_+(\boldsymbol{x})}{p_-(\boldsymbol{x})}\right)$ where $\gamma$ is any strictly increasing function. Due to the Neyman-Pearson lemma [27], $\mathrm{ROC}(t^*)$ has the highest TPR at any FPR level. Geometrically speaking, they dominate all other ROC cuves in an ROC plot and have the maximal AUC. Hence, we refer to $t^*$ as the optimal score and $\mathrm{ROC}(t^*)$ as the optimal ROC curve. For convenience, we denote the $\mathrm{ROC}(t^*)$ as $\mathrm{ROC}^*$ which reads "rock star". It can be shown that

$$\frac{f_+(t^*(\boldsymbol{x}_0))}{f_-(t^*(\boldsymbol{x}_0))} = \frac{\int_{\boldsymbol{x}:\gamma\left(\frac{p_+(\boldsymbol{x})}{p_-(\boldsymbol{x})}\right)=t^*(\boldsymbol{x}_0)} p_+(\boldsymbol{x})\mathrm{d}\boldsymbol{x}}{\int_{\boldsymbol{x}:\gamma\left(\frac{p_+(\boldsymbol{x})}{p_-(\boldsymbol{x})}\right)=t^*(\boldsymbol{x}_0)} p_-(\boldsymbol{x})\mathrm{d}\boldsymbol{x}} = \frac{p_+(\boldsymbol{x}_0)}{p_-(\boldsymbol{x}_0)}, \quad \forall\gamma \tag{3}$$

where the second equality holds due to $\frac{\int_D a(\boldsymbol{x})\mathrm{d}\boldsymbol{x}}{\int_D b(\boldsymbol{x})\mathrm{d}\boldsymbol{x}} = \gamma^{-1}(C)$, when $\gamma\left(\frac{a(\boldsymbol{x})}{b(\boldsymbol{x})}\right) \equiv C, \forall\boldsymbol{x} \in D$. When $\gamma(\boldsymbol{x}) = \boldsymbol{x}$, (3) expresses a known result [7], and is often given in plain English as "the density ratio of the likelihood ratio score is the likelihood ratio itself".

Finally, $\widehat{\mathrm{ROC}^*}$ takes an elegant form free from $t^*$ or $\gamma$: $\mathbb{E}_{p_-(\boldsymbol{x})}\sqrt{\left[\frac{p_+(\boldsymbol{x})}{p_-(\boldsymbol{x})}\right]^2+1}$. Equivalently,

$$\widehat{\mathrm{ROC}^*} - \sqrt{2} = \mathbb{E}_{p_-(\boldsymbol{x})}\left[g\left(\frac{p_+(\boldsymbol{x})}{p_-(\boldsymbol{x})}\right)\right] \tag{4}$$

We can see that the same $f$-divergence arises from computing the arc length of $\mathrm{ROC}^*$. However, unlike the $f$-divergence given in (2), (4) is an $f$-divergence between *data* distributions, not score distributions. It shows that as long as we use the the optimal scores, the arc length of the optimal ROC can indeed reflect the differences between *data* distributions. From now on, we will refer to $\widehat{\mathrm{ROC}^*} - \sqrt{2}$ as the ROC divergence. To the best of our knowledge, (4) has not been presented in literature before.

By definition, the ROC divergence is symmetric. Moreover, as a result of Proposition 1, the ROC divergence is upper bounded by $2 - \sqrt{2}$ and lower bounded by 0. Some geometric properties of $\mathrm{ROC}^*$ can be found in Section B.

## 4  Estimating the Arc Length of $\mathrm{ROC}^*$

### 4.1  A Variational Objective

To numerically approximate the arc length using samples alone, we leverage that $\widehat{\mathrm{ROC}^*} - \sqrt{2}$ is an $f$-divergence. Utilizing Fenchel's duality [20], authors in [29] show that an $f$-divergence $\mathrm{D}_g(p_+|p_-)$ has a variational representation:

$$\mathrm{D}_g(p_+|p_-) = \int_{\mathcal{X}} p_-(\boldsymbol{x})g\left[\frac{p_+(\boldsymbol{x})}{p_-(\boldsymbol{x})}\right]\mathrm{d}\boldsymbol{x} = \int_{\mathcal{X}} p_-(\boldsymbol{x})\sup_u\left\{u(\boldsymbol{x})\cdot\left[\frac{p_+(\boldsymbol{x})}{p_-(\boldsymbol{x})}\right] - g'[u(\boldsymbol{x})]\right\}\mathrm{d}\boldsymbol{x}$$

$$= \sup_u \int_{\mathcal{X}} p_+(\boldsymbol{x})u(\boldsymbol{x}) - \int_{\mathcal{X}} p_-(\boldsymbol{x})g'[u(\boldsymbol{x})]\mathrm{d}\boldsymbol{x},$$

where $g'$ is the convex conjugate of $g$ and the supremum is taken over all measurable functions. In the case of the ROC divergence, $g(z) = \sqrt{z^2+1} - \sqrt{2}$ and $z \in [0,\infty]$ thus $g$ has a convex conjugate $g'(z') = -\sqrt{1-z'^2} - \sqrt{2}$, $z' \in [0,1]$. Rewriting $\widehat{\mathrm{ROC}^*} - \sqrt{2}$ using the above variational representation and dropping the $-\sqrt{2}$, we obtain:

$$\widehat{\mathrm{ROC}^*} = \sup_{u\in[0,1]} \mathbb{E}_{p_+}[u(\boldsymbol{x})] + \mathbb{E}_{p_-}[\sqrt{1-u^2(\boldsymbol{x})}].$$

We reparameterize $u(\boldsymbol{x}) = \sin[v(\boldsymbol{x})]$, where $v \in [0, \pi/2]$:

$$\widehat{\mathrm{ROC}^*} = \sup_{v \in [0, \pi/2]} \mathbb{E}_{p_+} \sin[v(\boldsymbol{x})] + \mathbb{E}_{p_-} \cos[v(\boldsymbol{x})]. \qquad (5)$$

Differentiating the objective in (5) for $v$ and setting the derivative to zero, we can see the supremum is attained at $v^* = \operatorname{atan} \frac{p_+}{p_-}$. In other words, the optimal $v^*$ is the arctangent likelihood ratio function.

It is also interesting to see how $v^*$ is visualized in the ROC plot. We can see the tangent of $\mathrm{ROC}^*$ at an FPR level $s_0 \in [0, 1]$ is

$$\partial_s \tilde{F}_+(\tilde{F}_-^{-1}(s_0)) = \frac{f_+(\tilde{F}_-^{-1}(s_0))}{f_-(\tilde{F}_-^{-1}(s_0))} = \frac{p_+(\boldsymbol{x}_0)}{p_-(\boldsymbol{x}_0)}, \qquad (6)$$

where $\boldsymbol{x}_0$ is any point in $\mathcal{X}$ that satisfies the equality $\gamma\left(\frac{p_+(\boldsymbol{x}_0)}{p_-(\boldsymbol{x}_0)}\right) = \tilde{F}_-^{-1}(s_0)$. (6) is a known result [9]. In other words, $v^* = \operatorname{atan} \frac{p_+}{p_-}$ is the *slope angle* of $\mathrm{ROC}^*$. See Figure 2 for a visualization of the tangent of $\mathrm{ROC}^*$ expressed by the likelihood ratio.

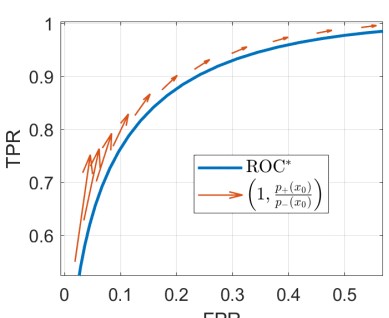

Figure 2: $\mathrm{ROC}^*$ and its tangent marked by vector $\left(1, \frac{p_+(\boldsymbol{x}_0)}{p_-(\boldsymbol{x}_0)}\right)$ (scaled to fit). $p_+ = \mathcal{N}(1, 1), p_- = \mathcal{N}(-1, 1)$.

Moreover, using (5), we can obtain a relationship between $\widehat{\mathrm{ROC}^*}$ and the total variation distance between $\mathbb{P}_+$ and $\mathbb{P}_-$ (denoted as $\mathrm{TV}(\mathbb{P}_+, \mathbb{P}_-)$).

**Proposition 2.** $\mathrm{TV}(\mathbb{P}_+, \mathbb{P}_-)$ *has a lower and upper bound expressed via* $\widehat{\mathrm{ROC}^*}$:

$$\max_{a \in [0,1]} \frac{2}{\pi} \left[ \frac{\widehat{\mathrm{ROC}^*} - 2\sqrt{1-a^2}}{a} + \arccos(a) - \arcsin(a) \right] \leq \mathrm{TV}(\mathbb{P}_+, \mathbb{P}_-) \leq \widehat{\mathrm{ROC}^*} - 1.$$

Proof of this proposition can be found in Section C. This proposition justifies that $\widehat{\mathrm{ROC}^*}$ is a valid measure of the discrepancy between $\mathbb{P}_+$ and $\mathbb{P}_-$. We compare this bound with other known TV bounds in Section 6.1.

## 4.2 A Tractable Objective for Estimating $\operatorname{atan} \frac{p_+}{p_-}$

To use (5) in practice, we need to find an appropriate function class $\mathcal{F}$. We can simply restrict $v$ to a bounded (parametric/non-parametric) function class $\mathcal{F}$ and solve the sample version of (5):

$$\max_{v \in [0, \pi/2], v \in \mathcal{F}} \frac{1}{n_+} \sum_{i=1}^{n_+} \sin(v(\boldsymbol{x}_i^+)) + \frac{1}{n_-} \sum_{i=1}^{n_-} \cos(v(\boldsymbol{x}_i^-)). \qquad (7)$$

In practice, enforcing the boundedness $v \in [0, \pi/2]$ over $\mathcal{X}$ is difficult. We can relax (7) by only enforcing the boundedness constraint of $v$ on the sample dataset $X_+ \cup X_-$.

For example, by letting $\mathcal{F}$ be a Reproducing Kernel Hilbert Space (RKHS) [33], we can translate (7) into the following optimization problem:

$$\hat{v} := \operatorname*{argmin}_{v \in \mathcal{H}} \ell(v) + \frac{\lambda}{2} \|v\|_{\mathcal{H}}^2, \quad \ell(v) := -\frac{1}{n_+} \sum_{i=1}^{n_+} \sin\langle v, \varphi(\boldsymbol{x}_i^+) \rangle - \frac{1}{n_-} \sum_{i=1}^{n_-} \cos\langle v, \varphi(\boldsymbol{x}_i^-) \rangle$$

$$\text{s.t: } \langle v, \varphi(\boldsymbol{x}) \rangle \in \left[0, \frac{\pi}{2}\right], \quad \forall \boldsymbol{x} \in X_+ \cup X_-, \qquad (8)$$

where $\mathcal{H}$ is a RKHS with a positive definite kernel $k(\boldsymbol{x}, \boldsymbol{x}') = \langle \varphi(\boldsymbol{x}), \varphi(\boldsymbol{x}') \rangle$, $\| \cdot \|_{\mathcal{H}}$ is the RKHS norm and $\frac{\lambda}{2} \|v\|_{\mathcal{H}}^2$ is the regularization term. The optimizer $\langle \hat{v}, \varphi(\boldsymbol{x}) \rangle$ is an estimation of $v^*(\boldsymbol{x})$, the arctangent of the likelihood ratio. (8) is a strictly convex optimization and thus, if a solution $\hat{v}$ exists, it must be unique.

Instead of modelling $\operatorname{atan} \frac{p_+}{p_-}$, we can opt for modelling the log likelihood ratio $\log \frac{p_+}{p_-}$. However, this modelling choice results in an non-convex optimization thus presents extra challenges in the theoretical analysis. Details can be found in Section K.

### 4.3 Finite Sample Guarantee

We show that the solution of (8), $\langle \hat{v}, \varphi(\boldsymbol{x}) \rangle$ converges to the true arctangent likelihood ratio (or its projection onto $\mathcal{H}$) as the number of samples goes to infinity. Below are a few regularity conditions.

**Assumption 1.** *There exists a unique $v^* \in \mathcal{H}$, such that $\mathbb{E}[\nabla_v \ell(v^*)] = 0$ and $\langle v^*, \varphi(\boldsymbol{x}) \rangle \in [0, \pi/2]$ holds for all $\boldsymbol{x} \in \mathcal{X}$.*

A sufficient condition of the above condition is specified in the following proposition.

**Proposition 3.** *If there exists a unique $v^* \in \mathcal{H}$, such that $\langle v^*, \varphi(\boldsymbol{x}) \rangle = \mathrm{atan}\left[\frac{p_+(\boldsymbol{x})}{p_-(\boldsymbol{x})}\right]$ then Assumption 1 holds.*

The proof can be found in Appendix F. Proposition 3 states if model is correctly specified and identifiable then Assumption 1 holds. It is possible that there exists a $v^* \in \mathcal{H}$ satisfying $\mathbb{E}[\nabla_v \ell(v^*)] = 0$ which does not meet the boundedness constraint $[0, \pi/2]$. In this paper we only consider situations where Assumption 1 holds, which includes all situations where the model is correctly specified and some situations where the model is misspecified.

**Assumption 2.** *Let $n_{\min} = \min(n_+, n_-)$. There exists a subspace $\mathcal{H}^* := \{v \in \mathcal{H} | \|v - v^*\|_{\mathcal{H}}^2 \leq \delta_{n_{\min}}^2\}$, such that $\forall v \in \mathcal{H}^*, \forall \boldsymbol{x} \in X_+ \cup X_-, \langle v, \varphi(\boldsymbol{x}) \rangle \in (0, \frac{\pi}{2})$ holds with high probability. The sequence $\delta_{n_{\min}}$ is monotonically decreasing as $n_{\min}$ grows to infinity.*

Assumption 2 states all $v$ within a vicinity of $v^*$ are in the interior of (8)'s feasible region with high probability. The following proposition gives a sufficient condition under which Assumption 2 holds.

**Proposition 4.** *Suppose $\|\varphi(\boldsymbol{x})\|_{\mathcal{H}} \leq 1$. If our model is correctly specified as described in Proposition 3 and $\forall \boldsymbol{x} \in \mathcal{X}, \mathrm{atan}\left[\frac{p_+(\boldsymbol{x})}{p_-(\boldsymbol{x})}\right] \in [R_1, R_2]$, for some $R_1$ and $R_2$ such that $\frac{\pi}{2} > R_2 > R_1 > 0$, then there exists $N > 0$ such that Assumption 2 holds when $n_{\min} > N$.*

The proof can be found in Appendix D. Since we use RKHS as the estimator function class, our final assumption is that $v^*$ should be reasonably smooth. In previous works such an assumption depends on the decay of the integral operator's eigenvalues [36, 10]. In this paper, we measure the smoothness using the *range space* technique which has been recently adopted in [11, 35]. We define

$$\boldsymbol{\Sigma}_v := \mathbb{E}[\nabla_v^2 \ell(v)] = \mathbb{E}_{p_+}[\sin\langle v, \varphi(\boldsymbol{x}) \rangle \cdot \varphi(\boldsymbol{x}, \cdot) \otimes \varphi(\boldsymbol{x}, \cdot)] + \mathbb{E}_{p_-}[\cos\langle v, \varphi(\boldsymbol{x}) \rangle \cdot \varphi(\boldsymbol{x}, \cdot) \otimes \varphi(\boldsymbol{x}, \cdot)],$$

where $\otimes$ denotes the outer product. Given $v_0 \in \mathcal{H}$, $\boldsymbol{\Sigma}_{v_0}$ is an integral operator on $u \in \mathcal{H}$ and

$$\boldsymbol{\Sigma}_{v_0} u = \mathbb{E}_{p_+}[\sin\langle v_0, \varphi(\boldsymbol{x}) \rangle \cdot \varphi(\boldsymbol{x}, \cdot) \cdot u(\boldsymbol{x})] + \mathbb{E}_{p_-}[\cos\langle v_0, \varphi(\boldsymbol{x}) \rangle \cdot \varphi(\boldsymbol{x}, \cdot) \cdot u(\boldsymbol{x})].$$

By definition $\boldsymbol{\Sigma}_{v_0}$ is a positive, self-adjoint operator, in the sense that $\langle u, \boldsymbol{\Sigma}_{v_0} u \rangle \geq 0$, $\langle u, \boldsymbol{\Sigma}_{v_0} v \rangle = \langle \boldsymbol{\Sigma}_{v_0}, u, v \rangle, \forall v, u \in \mathcal{H}$. Moreover, some algebra shows that $\boldsymbol{\Sigma}_{v_0}$ is also a bounded and compact operator. See Section G for more details.

Next, we assume the true arctangent ratio function (or its projection) is in the range space of $\boldsymbol{\Sigma}_{v^*}$.

**Assumption 3.** *Let $\mathcal{R}(\boldsymbol{\Sigma}_{v^*})$ denote the range space of $\boldsymbol{\Sigma}_{v^*}$. There exists $0 < \beta \leq 1$, $v^* \in \mathcal{R}(\boldsymbol{\Sigma}_{v^*}^\beta)$, where $C^\beta$ is the fraction power of a compact, positive and self-adjoint operator $C$.*

Note that the larger $\beta$ is, the smoother the functions in the range space are. More discussions on the range space assumption can be found in Section 4.2, [35]. Now we are ready to state our theorem:

**Theorem 1** (Convergence Rate of $\hat{v}$). *Suppose Assumptions 1, 2 and 3 hold and $\hat{v}$ exists. If $\|\varphi(\boldsymbol{x})\|_{\mathcal{H}} \leq 1$ and*

$$\lambda = \frac{T}{n_{\min}^{1/4}}, \quad \frac{K}{n_{\min}^{\beta/4}} \leq \delta_{n_{\min}} \leq \frac{4}{\max(B_+, B_-)},$$

*where $B_+ = \|(\boldsymbol{\Sigma}_{v^*} + \lambda \boldsymbol{I})^{-1} \mathbb{E}_{p_+}[\varphi(\boldsymbol{x})]\|_{\mathcal{H}}$, $B_- = \|(\boldsymbol{\Sigma}_{v^*} + \lambda \boldsymbol{I})^{-1} \mathbb{E}_{p_-}[\varphi(\boldsymbol{x})]\|_{\mathcal{H}}$ and $T \geq 1, K > 0$ are constants that do not depend on $n_{\min}$, then there exists a constant $N > 0$ such that when $n_{\min} > N$, $\|\hat{v} - v^*\|_{\mathcal{H}} = O_p(n_{\min}^{-\beta/4})$.*

The proof can be found at Appendix E. Theorem 1 shows that, under mild assumptions, $\hat{v}$ is indeed a good estimator for $\mathrm{atan}\left[\frac{p_+(\boldsymbol{x})}{p_-(\boldsymbol{x})}\right]$. Some discussions comparing our results with convergence results

proved by Nguyen et al. [28] can be found in Section M. Since $\mathrm{atan}\left[\frac{p_+(\boldsymbol{x})}{p_-(\boldsymbol{x})}\right]$ is an optimal score that gives rise to $\mathrm{ROC}^*$, our estimator may have some interesting applications such as outlier detection [19] or Neyman-Pearson classification [37]. We will defer discussions on those applications in future works. In the next section, we employ our arctangent likelihood ratio estimator in an application of lower bounding the maximal AUC.

## 5    Approximately Lower Bounding the Maximal AUC

Finding a score function $t$ that approximately maximizes AUC is an important task in binary classification. Let us denote the AUC of $\mathrm{ROC}(t)$ as $\mathrm{AUC}(t)$. It can be seen that

$$\mathrm{AUC}(t) = \int_{[0,1]} \tilde{F}_+(\tilde{F}_-^{-1}(s))\mathrm{d}s = \mathbb{E}_{p_-}\mathbb{E}_{p_+}\left[\mathbb{1}\left(t(\boldsymbol{x}_+) \geq t(\boldsymbol{x}_-)\right)\right].$$

Due to the Neyman-Pearson lemma, $\mathrm{ROC}^*$ has the maximum AUC among all ROC curves. Denote the AUC of $\mathrm{ROC}^*$ as $\mathrm{AUC}^*$ Consider the following inequalities:

$$\mathrm{AUC}^* = \underbrace{\sup_t \mathbb{E}_{p_-}\mathbb{E}_{p_+}\left[\mathbb{1}\left(t(\boldsymbol{x}^+) \geq t(\boldsymbol{x}^-)\right)\right]}_{(i)} \geq \sup_{t \in \mathcal{F}'} \underbrace{\mathbb{E}_{p_-}\mathbb{E}_{p_+}\left[L\left(t(\boldsymbol{x}^+), t(\boldsymbol{x}^-)\right)\right]}_{(ii)}, \qquad (9)$$

where $L(a, b)$ is a continuous and concave lower bound of the indicator function $\mathbb{1}(a > b)$. Due to the Neyman-Pearson lemma, the supremum of (i) is only attained when $t(\boldsymbol{x}) = \gamma\left(\frac{p_+(\boldsymbol{x})}{p_-(\boldsymbol{x})}\right)$ where $\gamma$ is a strictly increasing function. Replacing the expectations in (ii) with sample averages yields the optimization problem of AUC maximization [3, 12]:

$$\max_{t \in \mathcal{F}'} \frac{1}{n_-n_+} \sum_{i=1}^{n_-} \sum_{j=1}^{n_+} L(t(\boldsymbol{x}_j^+), t(\boldsymbol{x}_i^-)). \qquad (10)$$

The objective above is also referred to as Wilcoxon-Mann-Whitney statistic [16]. Therefore, we can see that AUC maximization is a procedure that approximates an optimal score function by maximizing a lower bound of $\mathrm{AUC}^*$.

Now, we show a different way of lower bounding $\mathrm{AUC}^*$ with the help of $\mathrm{atan}\,\frac{p_+(\boldsymbol{x})}{p_-(\boldsymbol{x})}$. We have seen that how $(\tilde{F}_-(\tau), \tilde{F}_+(\tau))$ parameterizes an ROC curve in Section 2.3. In fact, the area between $\mathrm{ROC}^*$ and the diagonal line from $(0,0)$ to $(1,1)$ can be similarly parameterized by considering ROC curves of positive and negative *mixture* score distributions.

Let $F_+^*$ and $F_-^*$ denote CDFs of any optimal score. For $\alpha \in [0, .5]$, we can define CDFs of $\alpha$-mixtures of $F_+^*$ and $F_-^*$ as follows

$$F_-^*(\tau, \alpha) := (1 - \alpha)F_-^*(\tau) + \alpha F_+^*(\tau),$$
$$F_+^*(\tau, \alpha) := \alpha F_-^*(\tau) + (1 - \alpha)F_+^*(\tau).$$

Then, FPR $(\tilde{F}_-^*(\tau, \alpha))$ and TPR $(\tilde{F}_+^*(\tau, \alpha))$ for these $\alpha$-mixtures can be defined accordingly. We visualize densities of $F_+^*(\alpha)$ and $F_-^*(\alpha)$ for different $\alpha$ in Figure 3.

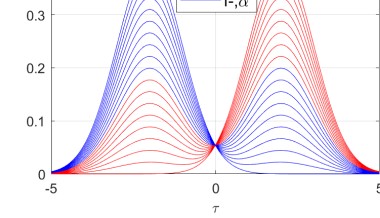

Figure 3:    Densities $f_+^*(\tau, \alpha)$ and $f_-^*(\tau, \alpha)$ for $\alpha \in [0, .5]$.

Further, we can see that the 2-dimensional coordinate $\boldsymbol{r}(\tau, \alpha) := (\tilde{F}_-^*(\tau, \alpha), \tilde{F}_+^*(\tau, \alpha))$ parameterizes the area between $\mathrm{ROC}^*$ and the diagonal line in $[0, 1]^2$:

- When fixing $\alpha$ and varying $\tau$, the coordinates give rise to a smooth curve in ROC space from $[0, 0]$ to $[1, 1]$.
    - When $\alpha = 0$, such a curve is $\mathrm{ROC}^*$.
    - When $\alpha = .5$, such a curve is the diagonal line.
- When fixing $\tau = \tau_0$ and varying $\alpha$, the coordinates produce a straight line segment connecting $(\tilde{F}_-^*(\tau_0, 0), \tilde{F}_+^*(\tau_0, 0))$ and $(\tilde{F}_-^*(\tau_0, .5), \tilde{F}_+^*(\tau_0, .5))$.

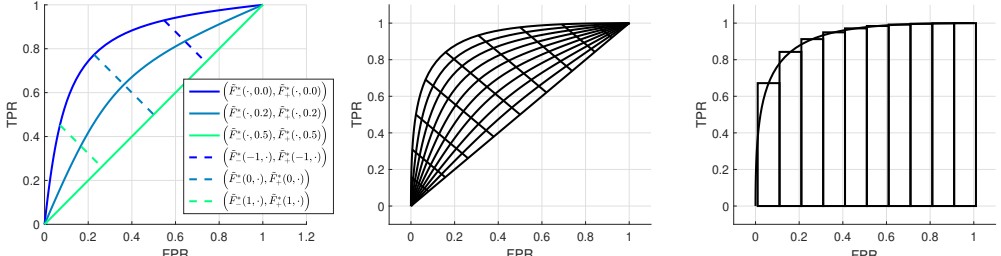

Figure 4: Left: $(\tilde{F}_-^*(\tau, \alpha), \tilde{F}_+^*(\tau, \alpha))$ parameterizes the surface between $\mathrm{ROC}^*$ and the diagonal line in $[0,1]^2$. This plot is created by setting $p_+ = \mathcal{N}(1, 1)$, $p_- = \mathcal{N}(-1, 1)$ and $t^*(x) = \frac{1}{2} \log \frac{p_+(x)}{p_-(x)} = x$. Middle: Our parameterization "mesh" divides AUC into surface elements (small patches on the plot). Right: Wilcoxon-Mann-Whitney statistic divides AUC into histogram bars.

The left plot in Figure 4 visualizes this parameterization. Now the surface area sandwiched between $\mathrm{ROC}^*$ and the diagonal line can be computed using a surface integration:

$$\mathrm{AUC}^* - .5 = \int_{\mathrm{dom}(\tau)} \int_{[0,.5]} \|\partial_\tau \boldsymbol{r}(\tau, \alpha) \times \partial_\alpha \boldsymbol{r}(\tau, \alpha)\| \, \mathrm{d}\alpha \mathrm{d}\tau, \tag{11}$$

where $\times$ denotes the cross product. After some algebra and applying the Fenchel duality technique in Section 4.1, we prove that $\mathrm{AUC}^*$ can be expressed as the supremum of a variational objective similar to (5):

**Proposition 5.** $\mathrm{AUC}^* = \frac{\sqrt{2}A}{2} + \frac{1}{2}$,

$$A := \sup_{v \in [0, \pi/2]} \mathbb{E}_{p_+} \left[ w \left( \mathrm{atan} \frac{p_+(\boldsymbol{x})}{p_-(\boldsymbol{x})} \right) \sin[v(\boldsymbol{x})] \right] + \mathbb{E}_{p_-} \left[ w \left( \mathrm{atan} \frac{p_+(\boldsymbol{x})}{p_-(\boldsymbol{x})} \right) \cos[v(\boldsymbol{x})] \right], \tag{12}$$

*where* $w(\tau) := \sin(\tau + \frac{\pi}{4}) \cdot |F_+^*(\tau) - F_-^*(\tau)|$. *The supremum of* (12) *is attained at* $v^* = \mathrm{atan} \frac{p_+}{p_-}$.

The proof can be found in Appendix H in the supplementary material. A lower bound of $A$ can be obtained by restricting $v$ to a function class.

Evaluating $w$ requires us to evaluate $\mathrm{atan} \frac{p_+(\boldsymbol{x})}{p_-(\boldsymbol{x})}$, $F_+^*$ and $F_-^*$ which are not readily available. However, Section 4.3 shows that the empirical estimator (8) is a consistent estimator of $\mathrm{atan} \frac{p_+(\boldsymbol{x})}{p_-(\boldsymbol{x})}$ under mild conditions. Therefore, we propose the following two-step procedure to approximately lowerbound $A$:

---

**Algorithm 1** Two-step Procedure for Approximately Lower Bounding $A$

---

1. Obtain $\hat{t}(\boldsymbol{x}) := \langle \hat{v}, \phi(\boldsymbol{x}) \rangle$ using (8). Approximate $F_+^*$ and $F_-^*$ using $\hat{F}_+$ and $\hat{F}_-$ which are empirical CDFs of $\hat{t}(\boldsymbol{x}^+)$ and $\hat{t}(\boldsymbol{x}^-)$.

2. Optimize the empirical version of (12) by restricting $v$ to a feasible function class and plugging in estimates obtained in the earlier step, i.e.,

$$\hat{v} := \operatorname*{argmax}_{v \in [0, \pi/2], v \in \mathcal{F}} \frac{1}{n_+} \sum_{i=1}^{n_+} \hat{w} \left[ \hat{t}(\boldsymbol{x}_i^+) \right] \cdot \sin[v(\boldsymbol{x}_i^+)] + \frac{1}{n_-} \sum_{i=1}^{n_-} \hat{w} \left[ \hat{t}(\boldsymbol{x}_i^-) \right] \cdot \cos[v(\boldsymbol{x}_i^-)], \tag{13}$$

where $\hat{w}(\tau) := \sin(\tau + \frac{\pi}{4}) \left| \hat{F}_+(\tau) - \hat{F}_-(\tau) \right|$.

---

Note that (13) is nothing but a weighted sample objective (7). Thus, it can be easily optimized by the algorithm that solves a weighted version of (7) given the approximated weights in the first step. In

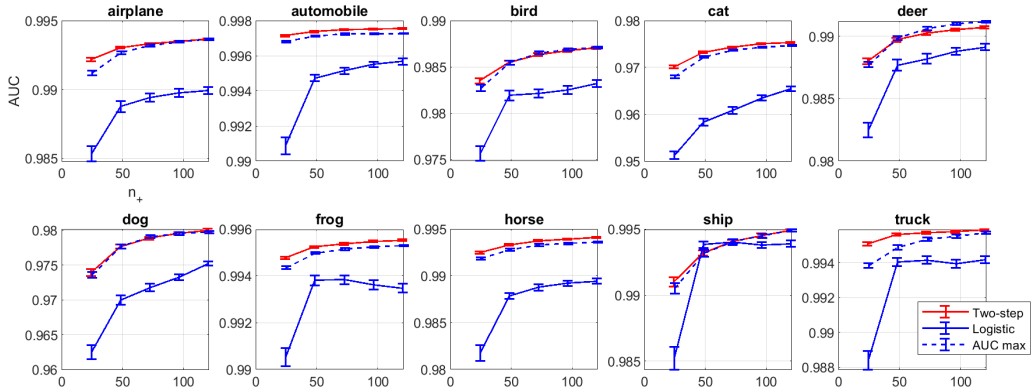

Figure 6: Testing AUC of one-versus-the-rest classification on CIFAR-10 dataset.

practice, we simply run the solver for (8) twice: The first time we run it without weights then run it again with weights $\hat{w}\left[\hat{t}(\boldsymbol{x}_i)\right]$ calculated from the first run.

Since the above algorithm also approximates an optimal score $(\operatorname{atan}\frac{p_+}{p_-})$ by maximizing an approximated lower bound of $\mathrm{AUC}^*$, it is natural to wonder how the maximizer $\hat{v}$ of (13) would perform in AUC maximization tasks. In the next section, we show that our two-step algorithm achieves a promising AUC performance compared to a state of the art AUC maximizer.

Computing $\mathrm{AUC}^*$ using (11) is different from using Wilcoxon-Mann-Whitney statistic (i.e., (13)): Our approach divides the space between $\mathrm{ROC}^*$ and the diagonal into small surface elements and then adds them up. Wilcoxon-Mann-Whitney statistic adds up all histogram bars, which are TPRs at different FPR levels. Our approach requires $\tilde{F}_+$ and $\tilde{F}_-$ to be differentiable with respect to $\tau$, which means the score distributions cannot be discrete. However, Wilcoxon-Mann-Whitney can compute the AUC of discrete score distributions without a problem. This difference is visualized in the middle and right plots of Figure 4.

## 6   Experiments

### 6.1   Numerical Comparison of Divergences and TV Bounds

In this experiment, we numerically compare the ROC divergence, the upper and lower bound in Proposition 2 with several other divergences and some known bounds of TV in Figure 5. In this numerical simulation, $p_+ = \mathcal{N}(0, 1)$ and $p_- = \mathcal{N}(\delta, 1)$. We plot ROC divergence, Jensen Shannon divergence, Wasserstein distance and TV between $p_+$ and $p_-$ as $\delta$ grows from 0 to 5. We can see the (rescaled) ROC divergence closely resembles TV. When $\delta > 1.45$, the upper bound given in Proposition 2 is the tightest among known TV upper bounds [2, 4] (Pinsker's upper-bound, Bretagnolle & Huber's upper bound, Le Cam's upper bound). This suggests that combining our upper-bound with existing bounds may produce an even tighter bound for TV.

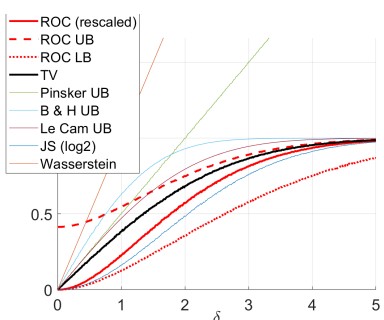

Figure 5: Comparison of various divergences and bounds of TV.

### 6.2   Imbalanced Classification on CIFAR-10

In this section, we test if the $\hat{\hat{v}}$ obtained in our two-step procedure (13) is indeed a good score function in terms of AUC in imbalanced classification tasks. We use a widely known image classification dataset CIFAR-10 [22]. The performance is compared with an AUC maximizer which maximizes the empirical lower bound in (10) and a vanilla logistic regression classifier. We set the surrogate loss $L(a, b) := -(1 - (a - b))^2$ in the AUC maximizer, as

suggested in [12]. All methods use linear models with no regularization terms since our models are simple and we have sufficient samples. Particularly, the $\hat{t}$ and $\hat{v}$ in our two-step algorithm is obtained using (8) by setting $\varphi(\boldsymbol{x}) = \boldsymbol{x}$ and $\lambda = 0$. The AUC maximization (AUC max) is implemented using SPAUC method [24].

Instead of using the raw features, we extract 50 dimensional bounded features by training a residual network [17] on the training dataset using the 10-class cross entropy loss. The structure of the network is included in the supplementary material. After obtaining features, we construct datasets for 10 different one-versus-the-rest classification tasks. For a single task, we pick a class and obtain $X_+$ by randomly sampling from this class $n_+$ times in the training set. Similarly, $X_-$ is obtained by randomly sampling from the rest of the classes $n_-$ times. In our experiments, we set $n_+ = 24, 48, 72, 96, 120$ and fix $n_- = 1000$ to create imbalanced positive and negative datasets. We run all three methods and obtain the corresponding score functions. For each class, we repeat the experiment 96 times using different random samples. We use the testing and training split provided by the dataset itself.

Our experiments can be seen as a transfer learning task which reuses predictive features trained for a multi-class classifier for one-versus-the-rest binary classification tasks.

The average AUCs computed on the testing dataset and their standard errors over 96 runs over different $n_+$ sample sizes are shown in Figure 6. Our method has approximately equal performance with the AUC maximizer despite not directly maximizing the AUC. This observation indicates that $\hat{v}$ can be a good score function in AUC maximization tasks. Both of the methods significantly outperform vanilla logistic regression.

## 6.3 Discussions on Computational Complexity

Without loss of generality, assume $n_+ = n_- = n$. The naive caclulation of objective (10) has a computational complexity $O(n^2)$ since we evaluate the loss function $L$ at each pair of samples. However, authors in [21] have shown that the objective function in (10) can also be computed with $O(n \log(n))$ complexity for hinge loss (and decomposable loss functions). A recent work [39] simplifies the computation of (10) for the squared loss function $L$ with an unbiased estimate. Suppose $t(\boldsymbol{x}) := \langle v, \boldsymbol{x} \rangle$, then the *negative* objective of (10) is an unbiased estimate of

$$1 + \mathrm{Var}_{p_+}[\langle v, \boldsymbol{x} \rangle] + \mathrm{Var}_{p_-}[\langle v, \boldsymbol{x} \rangle] + 2\langle v, \mathbb{E}_{p_-}[\boldsymbol{x}] - \mathbb{E}_{p_+}[\boldsymbol{x}] \rangle + \langle v, \mathbb{E}_{p_-}[\boldsymbol{x}] - \mathbb{E}_{p_+}[\boldsymbol{x}] \rangle^2. \quad (14)$$

After we approximating (14) with empirical terms, the computation can be done with a complexity $O(n)$. When implemented in an online fashion, it has a computational complexity of one datum.

In comparison, the objective (7) and (13) are summation of $\sin / \cos(v(\boldsymbol{x}))$ evaluated at each datum, so computing the objective/gradient has a computational complexity $O(n)$. Computing $\hat{F}_+$ and $\hat{F}_-$ requires sorting our dataset, which has an average complexity $O(n \log n)$. However, once our datasets are sorted, $\hat{F}_+ \left( \hat{t}(\boldsymbol{x}_0) \right) = \frac{i}{n_+}$, where $i$ is the index of $\hat{t}(\boldsymbol{x}_0)$ in the sorted set $\{\hat{t}(\boldsymbol{x}_i)\}_{i=1}^{n_+}$.

# 7 Conclusions

In this paper, we show that a novel $f$-divergence arises from the arc length of the optimal ROC curve. The arc length can be accurately estimated from positive and negative samples using a variational expression. It is also an estimator for $\mathrm{atan}\, p_+/p_-$ and has a convergence rate $O_p(n^{-\beta/4})$. Finally, we show that the area between the optimal ROC curve and the diagonal can be parameterized using a similar variational objective. It leads to a two-step procedure that approximately lower bounds the maximal AUC which achieves a promising result in AUC maximization tasks.

## Acknowledgments and Disclosure of Funding

The author would like to thank Prof. Peter Flach and Dr. Hao Song for their helpful discussions. The author would like to thank four anonymous reviewers for their insightful comments. In particular, we thank Reviewer WSBr and the Area Chair for pointing out the computational complexity inaccuracies in our initial version.

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
