# OpenReview forum: "Estimating the Arc Length of the Optimal ROC Curve and Lower Bounding the Maximal AUC"
_NeurIPS.cc/2022/Conference — NeurIPS 2022 Accept_

### Official Review · Reviewer_WSBr · 2022-07-08

**Rating:** 4
**Confidence:** 4
**Soundness:** 1 poor
**Presentation:** 3 good
**Contribution:** 2 fair

**Summary:**

This paper propose a novel method that utilizes arc length of the AUROC curve to approximate AUROC itself. Then, propose to optimize the arc length of the curve with some theoretical justifications for the relationship with AUROC.

**Questions:**

Please refer to the weakness part above. I like the new idea, but don't think it is ready to be published.

**Limitations:**

I don't see any limitations for societal impact.

**Strengths And Weaknesses:**

Strengths: The idea is novel and interesting. The paper provides some illustrations for the idea.

Weakness:

i) Optimizing arc length is not consistent with optimizing AUROC itself. As the examples shown in Figure 1, people could have the same optimized arc length, but have different optimized AUROC (subfigure a and b). Or, people could have the same optimized AUROC but have different optimized arc length (subfigure b and c).

ii) Although the idea is novel as an alternative way to optimizing AUROC, the paper lack some justifications for the advantages over the main-stream AUC optimization methods based on surrogate loss functions. (Gao, Wei, et al. "One-pass AUC optimization." International conference on machine learning. PMLR, 2013.;  Ying, Yiming, Longyin Wen, and Siwei Lyu. "Stochastic online AUC maximization." Advances in neural information processing systems 29 (2016).;  Liu, Mingrui, et al. "Fast stochastic AUC maximization with $ o (1/n) $-convergence rate." International Conference on Machine Learning. PMLR, 2018. etc.)

iii) The experiments are insufficient. Only CIFAR10 is conducted for the experiment. And only logistic regression and a pairwise squared loss optimization method are considered as comparisons. There is no deep learning experiments. Moreover, even based on the current experimental results, the proposed method doesn't outperform the pairwise loss optimization.

iv) There are some short claims for the computational efficiency of proposed method, but there is no justifications.

---

> ### Author Response · Authors · 2022-08-02
> **Respond and Clarifications**
>
> > Then, propose to optimize the arc length of the curve with some theoretical justifications for the relationship with AUROC...
> > Optimizing arc length is not consistent with optimizing AUROC itself.
>
> This seems to be a misunderstanding: Nowhere in this paper have we advocated optimizing arc length. We propose to "estimate the arc length of the optimal ROC curve", which is markedly different from optimizing the arc length.
>
> Could the reviewer please point out the sentences/paragraphs that may have lead to this misunderstanding?
>
> > the paper lack some justifications for the advantages over the main-stream AUC optimization methods based on surrogate loss functions.
>
> This seems to be another misunderstanding. The mentioned papers solve the *online/stochastic* AUC maximization. Online/stochastic algorithms have smaller computational/memory complexities but will always converge more slowly than their offline counterparts. These algorithms are only used when the dataset does not fit into the memory. The offline algorithm will perform best if the computation can entirely fit into the memory.
>
> In other words, replacing our offline algorithm with any of the mentioned algorithms will not result in better performance for the AUC maximizer in our experiments. We do not compare with these algorithms as the proposed method works in an offline setting. We added clarifications about this point in the uploaded revision.
>
> > the proposed method doesn't outperform the pairwise loss optimization.
>
> We do not expect the proposed method to perform better. Since both methods maximize a lowerbound of maximal AUC, they should perform comparably. However, the proposed method has a lower computational complexity in the *offline setting*.
>
> > There are some short claims for the computational efficiency of proposed method, but there is no justifications.
>
> We have made 2 claims of the computational complexities. (12) (or (11) in the revision) has a complexity O(n). Computing F_hat has a complexity O(nlogn).
>
> The objective (12) is a summation of sin/cos(v(x)) evaluated at each datum, so computing the objective/gradient has a computational complexity O(n).
>
> Computing F_hat requires sorting. Popular methods (such as quicksort or mergesort) have an average computational complexity O(nlogn).
>
> Since our algorithm computes F_hat in the first step, then plugs the already computed F_hat into the second step, our algorithm will have a computational complexity O(nlogn) in the first step, and O(n) in the second step.
>
> Some arguments along these lines can be found on line 280 and 281. We added some further explanations in the revision.

---

> > ### Comment · Reviewer_WSBr · 2022-08-03
> > **Follow up with authors responses.**
> >
> > **Thanks to authors' responses and endeavors for the work, which could improve the paper. However, based on the responses, I still think the contribution is limited for acceptance; therefore, I decide to keep the same score.**
> >
> > >  This seems to be a misunderstanding: Nowhere in this paper have we advocated optimizing arc length. We propose to "estimate the arc length of the optimal ROC curve", which is markedly different from optimizing the arc length.
> >
> > The core object function at the paper, equation 7, basically optimizes the model parameters to maximize arc length. The 'estimate' authors mentioned could also be regarded as 'optimize'.
> >
> > One key for this part, the objective is not consistent with maximizing AUROC, as the question I raised at the beginning. If it is not consistent, how could we utilize the method in practice applications without any worries?
> >
> > > This seems to be another misunderstanding. The mentioned papers solve the online/stochastic AUC maximization...
> >
> > As we have entered the big data era, online/stochastic methods is essential for the machine learning applications. Thanks for taking my concern and adding the stress point to the revised version of the paper, it is good to let audience know the limitations.
> >
> > > We do not expect the proposed method to perform better. Since both methods maximize a lowerbound of maximal AUC, they should perform comparably. However, the proposed method has a lower computational complexity in the offline setting.
> > > We have made 2 claims of the computational complexities. (12) (or (11) in the revision) has a complexity O(n). Computing F_hat has a complexity O(nlogn).
> >
> > Thanks for adding the statements for those parts. Yes, I agree with the point that pairwise loss functions take O(n^2) for offline setting, but as we discussed before, we could use stochastic algorithm for them and they are consistent with maximizing AUROC. Furthermore, for composite loss function (Ying, Yiming, Longyin Wen, and Siwei Lyu. "Stochastic online AUC maximization." Advances in neural information processing systems 29 (2016)), it only takes O(n) for offline setting and is consistent with maximizing AUROC.
> >
> >
> > The experiments (one dataset with two basic anchor baselines) are also not sufficient for justifying the proposed method.
> >
> >
> > **Although the idea for this paper is interesting, I still think it is below the acceptance for lack of theoretical and experimental justification for AUROC maximization application.**

---

> > > ### Author Response · Authors · 2022-08-05
> > > **Some Further Clarifications**
> > >
> > > Thanks for your prompt reply! Some further clarifications:
> > >
> > > > The 'estimate' authors mentioned could also be regarded as 'optimize'.
> > >
> > > No. The $\max$ in objective (7) does *not* mean maximizing the arc length. The maximization in (7) comes from the $\sup$ in (5).
> > > The sup in (5) comes from the application of [Fenchel duality](https://en.wikipedia.org/wiki/Fenchel%27s_duality_theorem):
> > >
> > > For any convex function $g$, Fenchel's duality states:
> > > $g(z) = \sup_u uz - g'(u)$, where $g'$ is the convex conjugate of $g$. The equality holds at the supremum, hence the $\sup$ in (5).
> > > In our case, $g(z) = \sqrt{z^2 + 1} - \sqrt{2}$. Please see Section 4.1 for more details.
> > >
> > > To sum up, the $\sup$ in (5) (and $\max$ in (7)) is a way of approximating the arc length and is **not** an attempt to maximize arc length. This technique has been used to approximate many other $f$-divergences, e.g., (4) in [Nowozin et al., 2016](https://arxiv.org/pdf/1606.00709.pdf).
> > >
> > > We totally understand the reviewer's comments that optimizing arc length $\neq$ optimizing AUC, but we really do not optimize the arc length in this work. Hence this inconsistency does not affect the utility of our method. In fact, we know optimizing arc length is not consistent with maximizing AUC, so we wrote Section 5 to expressed the maximal AUC as a variatoinal objective and lower bound it using a two-step Algorithm.
> > >
> > > > it only takes O(n) for offline setting and is consistent with maximizing AUROC.
> > >
> > > We understand the reviewer is concerned about us not comparing with online/stochastic AUC maximizers. However, the main contribution of our paper is not about imbalanced classification, but to provide novel expressions for some fundamental ROC geometries. Section 6 is only to highlight the **potential** of our optimal AUC variational expression and the arc length estimator in imbalanced classification.
> > >
> > > In fact, developing a stochastic/online algorithm itself is significant enough to warrant a new paper (all papers mentioned by the reviewer belongs to this category). Given other novel works that have already been included in this paper, studying our new objective under a online/stochastic setting will be a future work.
> > >
> > > We will state in our paper that:
> > >
> > > "Note that the state of the art algorithms that maximizes Wilcoxon-Mann-Whitney statistics do not have to solve O(n^2) optimization in online/stochastic settings (such as SOLAM proposed by Ying et al., 2016). However, developing such online/stochastic algorithms for our objective (12) will be a future task."

---

> > > > ### Author Response · Authors · 2022-08-08
> > > > **Results Update**
> > > >
> > > > We have updated the results in our imbalanced classification experiments. Now the AUC maximization (AUC max) is implemented using SPAUC method (Lei and Ying, 2021), which extends Ying et al., 2016.
> > > >
> > > > As we predicted in our previous post, the result is almost identical since our original implementation optimizes the same surrogate loss (eq. 8 in the revision) as the stochastic algorithms such as SPAUC. We updated the code in the supplementary material too.
> > > >
> > > > However, we stress that all methods mentioned by the reviewer work in a very specific setting: **Dataset comes in as a stream, and the optimization is only carried out one datum at a time without accessing the previous data points**. This setting is very specialized and targeted, and a comparison with works along this line should be future work.

---

> > > > > ### Comment · Reviewer_WSBr · 2022-08-09
> > > > > **Follow up with the feedbacks**
> > > > >
> > > > > > The related work (Ying et al., 2016.) is for online learning.
> > > > >
> > > > > Yes, it is proposed for online learning. But **it also could be used for offline setting**. As the author claimed that their proposed method has **O(nlog(n))** complexity for offline setting, but the method (Ying et al., 2016.) **only requires O(n)**. For this comparison, I am **not** referring any stochastic/online gradient method analysis. Please refer to the theorem 1 of (Ying et al., 2016.). The pairwise loss could be decoupled and re-arranged to linear form.
> > > > >
> > > > > If there is no computational advantage for the proposed method, I would concern the limited contribution for this paper. There is no demonstration for the superiority from experiments as well.
> > > > >
> > > > > > The paper is not optimizing arc length.
> > > > >
> > > > > The authors design their method as a two-step method, where the arc length is the key 'anchor' point at the first step for their method. I am not convinced that their final objective for the proposed method is consistent with optimizing the AUROC, given that first step doesn't necessary provide the ideal referenced maximal AUROC curve.
> > > > >
> > > > > I agree that the proposed method could optimize the lower bound of the AUROC though (section 5). But what's the advantage for this approach comparing to the classical surrogate loss methods?

---

> > > > > > ### Author Response · Authors · 2022-08-09
> > > > > > **Further Reply**
> > > > > >
> > > > > > > The pairwise loss could be decoupled and re-arranged to linear form.
> > > > > >
> > > > > > Thanks for pointing it out. Please allow us to clarify further.
> > > > > >
> > > > > > The result in Theorem 1 Ying et al., 2016 depends on two things:
> > > > > >
> > > > > > 1. Squared surrogate loss $(1-w(x-x'))^2$
> > > > > > 2. Linear model $t(\boldsymbol{x}) = \boldsymbol{w}^\top \boldsymbol{x}$
> > > > > >
> > > > > > For example, if the score function $t(\boldsymbol{x})$ is a deep neural network model, the decoupling result in Theorem 1 does not hold. As a result, one has to use a pairwise loss function with computational complexity O(n^2).
> > > > > >
> > > > > > In comparison, the proposed AUC parameterization does not rely on
> > > > > > 1. Any surrogate loss (Proposition 6).
> > > > > > 2. Any specific parameterization of the model.
> > > > > >
> > > > > > For example, if we replace the model family $\mathcal{F}$ in (12) with a deep neural network model, the objective function still enjoys O(n) complexity.
> > > > > > We have added relevant discussions in the revision (lines 267-269), highlighting the decoupling technique proposed by Ying et al., 2016 and its potential limitations.
> > > > > >
> > > > > > > given that first step doesn't necessary provide the ideal referenced maximal AUROC curve.
> > > > > >
> > > > > > Theorem 1 shows that the first-step estimator converges to arctan(p+/p-), the optimal testing score.
> > > > > >
> > > > > > Therefore, if the first step supplies a good plug-in estimate of arctan(p+/p-), the second step should approximate the AUC nicely (Proposition 6).
> > > > > >
> > > > > > We added a simulation in Section K (appendix) to demonstrate the estimation precision of the first step estimator. The result shows, our estimator derived from the ROC-divergence can indeed approximate the true arctan(p+/p-) well.

---

### Official Review · Reviewer_RsuF · 2022-07-13

**Rating:** 6
**Confidence:** 3
**Soundness:** 3 good
**Presentation:** 4 excellent
**Contribution:** 3 good

**Summary:**

This paper builds a connection between the ROC curve of testing two distribution $p_{+}(\boldsymbol{x})$,  $p_{-}(\boldsymbol{x})$ and the "distance" between these two distributions. In particular, the length of the optimal ROC curve corresponds to a type of $f$-divergence between $p_{+}(\boldsymbol{x})$ and $p_{-}(\boldsymbol{x})$. This $f$-divergence can be estimated utilizing its variational formulation. The optimal solution of the equivalent variational problem is $\arctan[\frac{p_{+}(\boldsymbol{x})}{p_{-}(\boldsymbol{x})}]$. Therefore, by solving the empirical version of this variational problem, an estimate of $\arctan[\frac{p_{+}(\boldsymbol{x})}{p_{-}(\boldsymbol{x})}]$ can be obtained.

**Questions:**

1. For the two-step procedure stated in Line 264, why do we need the second step? The goal is to estimate $\arctan[\frac{p_{+}(\boldsymbol{x})}{p_{-}(\boldsymbol{x})}]$, which has been done in Step 1.

2. It turns out that the optimal solution of the variational problem (Line 141) is a monotone transform of likelihood function. Can the author comment on under which choice of $g$, this is the case?

3. Is there a typo in Eq. (1): Should it be $[\partial_{\tau}\tilde{F}_{-}(\tau)]^2$ ?


**Strengths And Weaknesses:**

Strengths: The fact that the length of optimal ROC curve corresponds to a $f$-divergence is an interesting new result. Also the presentation of this paper is clear and fun to read.
Weakness: I think the condition in Proposition 4, which requires that $\arctan[\frac{p_{+}(\boldsymbol{x})}{p_{-}(\boldsymbol{x})}] \in \mathcal{H}$ is a little stringent. In practice, to reduce computational complexity, the class $\mathcal{H}$ may not be very rich. It is better to discuss what will happen when there exist some approximation error of $\arctan[\frac{p_{+}(\boldsymbol{x})}{p_{-}(\boldsymbol{x})}]$. Also I think it will be better if more numerical experiments are provided, e.g., comparison with other methods of choosing score function $t(x)$ (if any) and experiments in other datasets.

---

> ### Author Response · Authors · 2022-08-02
> **Respond and Clarifications**
>
> > the condition in Proposition 4, which requires that atan p_+/p_- in H is a little stringent.
>
> Proposition 4 is only a *sufficient* condition under which Assumption 1 holds. Assumption 1 does cover cases where atan p_+/p_- is not in $\mathcal{H}$ as we explained in the paragraph following Proposition 4. When our model is misspecified, the estimate converges to the projection on $\mathcal{H}$ (v* defined in Assumption 1).
>
> > For the two-step procedure stated in Line 264, why do we need the second step?
>
> Yes, the first step does provide *an* estimate of atan p+/p-. The second step produces a different estimate by maximizing the AUC. Therefore, we expect the estimate from the second step to perform well in AUC maximization tasks (Section 6). The same cannot be said for the first step estimate, as it is not obtained by maximizing the AUC.
>
> > Can the author comment on under which choice of g, this is the case?
>
> In line 143: g(z) = sqrt(z^2 + 1) - sqrt(2).
>
> > Is there a typo in Eq. (1):
>
> Thank you for pointing it out. It has been fixed in our uploaded revision.

---

### Official Review · Reviewer_zcmh · 2022-07-13

**Rating:** 7
**Confidence:** 4
**Soundness:** 4 excellent
**Presentation:** 4 excellent
**Contribution:** 4 excellent

**Summary:**

This paper discusses f-divergence with arc length of ROC curve.  The authors find that several papers use ROC curves to compare two distributions.  And then try to answer the reason why arc length of ROC curve could be a type of f-divergence. The authors build the connection between the arc length of ROC curve and f-divergence, and then derive the algorithm to estimate the arc length of ROC curve. The experiments on Cifar-10 show that the score function obtained from 2-step procedure produced promising results.

**Questions:**

It would be great if the author could address the questions in weakness and comments.

**Strengths And Weaknesses:**

Strength:

1. The paper provides a good presentation to the paper.

2. The paper starts with a question of using ROC curves to compare two distributions, and then answers this question step by step. The deviations in the main body make sense.


Weakness:
1. Although the authors have convinced me that the arc length of ROC curve is a type of f-divergence similar to KL divergence, total variational distance, the relationship with them are not discussed well. I only find proposition 3 discuss the relationship with total variational distance. More relations are appreciated to be provided. In addition, in the experiments, the authors only show the rationality of the arc length of ROC curve, the discussion and comparison with other f-divergences are missed here.

2. Although the paper is of a good presentation, still has some minor types, like line 30 the font of f should be mathematical way, line 32, there has one more space between [28] and ‘.’.

---

> ### Author Response · Authors · 2022-08-02
> **Respond and Clarifications**
>
> > the relationship with other f-divergence & the discussion and comparison with other f-divergences are missed here.
>
> We have added a comparison plot (Figure 3) in the uploaded revision, comparing ROC divergence, TV distance, Wasserstein distance, Jensen-Shannon divergence, and several known upper/lower bounds of TV distance (including the ones we derived in Proposition 3).
>
> Compared with other known TV upper bounds, ROC upper bound is tighter in our numerical example when two distributions are further apart.
>
> Please see Section 6 for more details.
>
> > Typos on line 30 and 32.
>
> Thanks for pointing them out. They have been fixed in the uploaded revision. We are going to check for all typos in the future revision thoroughly.

---

### Official Review · Reviewer_2jFB · 2022-07-18

**Rating:** 6
**Confidence:** 3
**Soundness:** 3 good
**Presentation:** 2 fair
**Contribution:** 3 good

**Summary:**

f-divergences can be written as variational objectives, typically parameterised in terms of a density-ratio estimator between 'positive' and 'negative' distributions. This paper identifies a new equivalence between a particular kind of f-divergence and the arc length of the `optimal AUC curve' (i.e. the best possible AUC curve amongst all classification scoring-functions). Convergence guarantees are given when the  estimated (arctan) log-density-ratio belongs to an RKHS. Finally, this estimator is used to (approximately) lower bound the optimal AUC, which is an important quantity in unbalanced binary classification settings, as shown in the experiments on Cifar-10.

**Questions:**

Could the authors comment on the added value of their theoretical convergence results over that of Nguyen et al. ? (2007, "Estimating divergence functionals and the likelihood ratio by penalized convex risk minimization). A superficial glance at that paper suggests that they prove a rather similar result for *arbitrary* f-divergences.

If I am correct that your results are similar to the prior work (but perhaps involve small modifications), then this should be made clear in the exposition.

Right now, I have assigned a score of 'fair' to the contributions of this paper. I would be willing to raise this if these concerns regarding theoretical novelty turn out to be wrong.


Minor points

line 40: the term 'surface area' is unclear here.  How does it differ from the usual AUC? needs clarifiying.
line 73-74: Readers may be used to thinking of FPR & TPR as fixed scalars (rather than functions of a threshold). I think it could help to give these functions arguments and perhaps refer to such arguments as `thresholds'.

minor correction: line 40: `maximual' --> `maximal'

**Limitations:**

The authors are very upfront about certain limitations (e.g. the fact that they require differentiabiltity of the score function w.r.t its input, unlike prior work).

One issue is that their `lower bound' on the AUC does not appear to truly be a lower-bound (since they have to replace certain terms with approximations). This limitation could be made clearer, since right now, the conclusion implies otherwise.

**Strengths And Weaknesses:**

Strengths

- It's an interesting, non-obvious finding that the ROC curve can be connected to variational f-divergence representations. Also, it takes some ingenuity to consider the arc-length rather than the more standard AUC.

- As far as I can tell, the theoretical results look correct (although I have concerns about novelty - see below)

- The experiment is interesting and shows some promise (but I have concerns - see below).

Weaknesses

- I have questions about the novelty of the theoretical results. See Questions below.

- I think it is *very* important to report the wallclock times for the different methods used in the experiments. I understand that the proposed method has better asymptotic complexity, but I want a sense of how big those hidden constant factors are.

- Section 4.1 (line 146 onwards) & Section 4.2 were hard to follow. In equation 5, the variational formula is reparameterised in term of an angle v \in [0, \pi/2]. At this point in the exposition, it is quite unclear why such a reparameterisation is a good idea; it is more standard to parameterise the variational formla in terms of the log-density-ratio, which belongs to [0, \infty]. A whole page later, we discover that the log-density-ratio approach (combined with a linearity assumption) is theoretically less convenient because of nonconvexity. I think that this nonconvexity issue could be mentioned *before* Equation 5, so the reader understands where the argument is going.

- Section 5 (lines 232-256) is very math-notation-heavy, making it difficult to read. More plain-english explanations would greatly improve clarity. For instance, many explanations could be stated in terms of the lines in Figure 2, rather than using math.

- the abstract is rather vague, and I struggled to understand it (In contrast, the introduction was quite clear). I recommend clarifying what is meant by `score function', since it has multiple meanings (e.g. the gradient of log-likelihood w.r.t parameters/data).

---

> ### Author Response · Authors · 2022-08-02
> **Respond and Clarifications**
>
> > Could authors comment on the added value of their theoretical convergence results over that of Nguyen et al. ? (2007, "Estimating divergence functionals and the likelihood ratio by penalized convex risk minimization).
>
> The convergence results in Nguyen et al. 2007 are only developed for two estimators derived from **KL divergence**, so they are **not** general theories for arbitrary f divergences. Their proofs cannot be easily applied to our ROC divergence.
>
> Moreover, Theorem 1 is not a minor modification of convergence theories in Nguyen et al. 2007. Specifically, Nguyen et al. 2007 prove the likelihood ratio converges in *Hellinger distance*, while we prove the *arctangent likelihood ratio* converges in Hilbert space norm. The proofs rely on completely different machinery and assumptions. These technical results depend on the variational objective functions the estimators maximize and are not interchangeable.
>
> We are more than happy to expand on this issue further. Please do let us know if you would like a more technical response.
>
> > it is very important to report the wallclock times for the different methods used in the experiments.
>
> We have added a wall clock time comparison between the proposed method and AUC maximization in the uploaded revision. Both methods are implemented using MATLAB's optimization toolbox. The code is in the updated supplementary material.
>
> We can see that the computation time of the proposed method grows at a much slower rate than the AUC maximization.
>
> > Confusing statements at Line 40 and 73-74... the abstract is rather vague...
>
> Thanks for pointing them out, and we revised these statements and rewritten the abstract in the uploaded revision.

---

> > ### Comment · Reviewer_2jFB · 2022-08-08
> > **Thank you for the clarifications. I've increased my score.**
> >
> > Thank you for clarifying the differences between your work & that of Nguyen et al. (2007). I think that you should include this comparative discussion somewhere in the text (appendix is fine). I've increased my score based on this clarification. I now rate the contribution as 'good' and recommend weak accept.

---

> > > ### Author Response · Authors · 2022-08-09
> > > **Thank you!**
> > >
> > > Thank you for increasing the score! We will include the comparative discussion in the revision!

---

### Author Response · Authors · 2022-08-02
**Thank You for Insightful Comments!**

We thank all reviewers for their insightful feedback. Please find our responses to your comments below. We revised our paper and added two new results that helped us respond to reviewers' comments:

- Figure 4: The wall clock time comparison between the AUC maximization and the proposed two-step procedure in **the offline setting**.
- Figure 3: A numerical comparison between ROC divergence and various other divergence and bounds of TV distance (including ones derived in Proposition 3).

All revised paragraphs are highlighted in red. We are happy to discuss with reviewers if they have further questions.

---

### Author Response · Authors · 2022-08-09
**Thank You and Concluding Remark!**

During the discussion, Reviewer WSBr raised an interesting point that Theorem 1 in Ying et al., 2016 offers a way of rewriting the pairwise AUC loss into a linear form, which has a computational complexity of O(n), instead of O(n^2).

We would like to comment that this technique only works for linear models ($t(x) = wx$) and the squared surrogate loss. Thus, this reformulation does not apply when, e.g. the model is a neural network.

In comparison, our parameterization does not depend on any surrogate loss and specific model family. Objective (11) will always have a computational complexity O(n) regardless of the model choice. We have commented on this in the revision, line 269-271. More details can be found in our discussion with Reviewer WSBr.

We sincerely hope that our clarifications can be taken into account during the next phase of the discussion. We thank all reviewers for their insightful comments.

Y. Ying, L. Wen, and S. Lyu. Stochastic online auc maximization. In Advances in Neural Information Processing Systems 29 (NeurIPS 2016), 2016

---

### Meta-Review · Area_Chair_n9aa · 2022-08-27

**Recommendation:** Accept
**Confidence:** Less certain

**Metareview:**

The paper shows that the arc length of the optimal ROC curve is an $f$-divergence, propose san estimator for it, and builds on the insights obtained to design a new algorithm for approximately maximizing the area under the ROC curve.

The reviewers generally appreciate the theoretical results / insights. The only concern seems to be about the empirical evaluation of the AUC maximization procedure and about the lack of sufficient comparison to the state-of-the-art AUC maximization methods.

Given that the main contribution is the connection drawn between the arc length of ROC curve and $f$-divergence, a majority of the reviewers are in favor of accepting the paper even if the empirical evaluation is not entirely satisfactory.

**Authors inaccurate about offline AUC maximization taking $O(n^2)$ time**

One of the selling points in the paper is that the new AUC maximization approach achieves a better run-time complexity than more traditional methods for AUC-maximization. Based on the authors' back-and-forth with Reviewer WsBr, it appears that the method of Ying et al. (2016) already achieves an O(n) run-time even for the offline setting.

I would also like to point to the authors that, dating back to as early as 2005, there have been methods for offline AUC maximization with O(n log(n)) computational cost. For example, with the pairwise SVM loss, Joachims (2005, lemma 2) show that the loss/gradient computation only requires $O(n \log(n))$ computation. This computational time applies to both linear and non-linear models, as I elaborate below.

Suppose there are $n^+$ positive examples and $n^-$ negative examples, and we would like to minimize the pairwise hinge loss for scoring function $f$:

$L(f) = \sum_{i=1}^{n^+} \sum_{j=1}^{n^-} \phi(f(x^+_i) - f(x^-_j) )$

where $\phi(z) = \max(0, 1 - z)$. Computing this loss does not need us to explicitly enumerate $O(n^2)$ pairs. Instead it suffices to sort the positives according to $f(x^+_i)$ and the negatives according to $1 + f(x^-_j)$, and then compute the following cumulative stats by taking a single pass (O(n)) over the sorted examples:

$N_{i}^{+}= \sum^{n^-}_{j=1} \mathbb{I}( 1 + f(x^-_j) \geq f(x^+_i)  )$

$L_{i}^{+}= \sum_{j:~ 1 + f(x^-_j) \geq f(x^+_i)  } f(x^-_j)$

The pairwise loss can then be computed in O(n) time:

$L(f) = \sum_{i=1}^{n^+} L_{i}^{+} + N_{i}^{+} \cdot (1 - f(x^+_i) )$

A similar procedure can be used to compute gradients for the pairwise loss, and would again require only $O(n\log(n))$ computation (for the sorting step).

*Ref*: Joachims, A Support Vector Method for Multivariate Performance Measures, ICML 2005.

**Recommended changes/inclusions to camera-ready version**

We are accepting this paper under the expectation that the authors will include a more accurate description off-line AUC maximization methods, and accurately describe the exact computational advantages their method has (if any) over prior AUC maximization methods. If there are none, please don't highlight them in the paper. A review of stochastic AUC maximization methods is also highly desirable.

**Award:**

No

---

### Decision · Program_Chairs · 2022-09-14

Accept